

# Isolation and identification of a halophilic and alkaliphilic microalgal strain

Chenxi Liu, Jiali Liu, Songmiao Hu, Xin Wang, Xuhui Wang and Qingjie Guan

Key Laboratory of Saline–alkali Vegetation Ecology Restoration (SAVER),
Ministry of Education, Alkali Soil Natural Environmental Science Center (ASNESC),
Northeast Forestry University, Harbin, China

## ABSTRACT

Halophilic and alkaliphilic microalgal strain SAE1 was isolated from the saline–alkaline soil of Songnen Plain of Northeast China. Morphological observation revealed that SAE1 has a simple cellular structure, single cell, spherical, diameter of four to six μm, cell wall of about 0.22 μm thick, two chloroplasts and one nucleus. Analysis of the phylogenetic tree constructed by 18S sequence homology suggests that SAE1 is highly homologous to *Nannochloris* sp. *BLD-15*, with only four base substitutions in the homologous region. SAE1 was initially considered as *Nannochloris* sp. Analysis of the halophilic and alkaliphilic characteristics of SAE1 indicates that it can grow under one M $NaHCO_3$ and NaCl concentrations, with optimal growth under 400 mM $NaHCO_3$ and 200 mM NaCl. The intracellular ultrastructure of SAE1 significantly changed after NaCl and $NaHCO_3$ treatments. A large number of starch grains accumulated after treatment with 400 mM $NaHCO_3$ in cells, but few were found after treatment with 200 mM NaCl and none in the living condition without treatment. We conjectured that one of the metabolic characteristics of alkaliphilic ($NaHCO_3$) microalga SAE1 is the formation of massive starch grains, which induce glycerol anabolism and increase osmotic pressure, thereby enhancing its ability to resist saline–sodic conditions. This feature of alkaliphilic ($NaHCO_3$) microalga SAE1 contributes to its growth in the carbonate soil of Songnen Plain.

## INTRODUCTION

Current soil environment is threatened by salinization (*Ruppel, Franken & Witzel, 2013*; *Navarro-Torre et al., 2017*). Introducing heterologous genes to plants during cultivation is a new strategy to adapt to extreme environments, such as saline–alkali environments. Isolating germplasm or genetic resources is the primary prerequisite in this technique (*Ohki et al., 2011*; *Wei, Takano & Liu, 2012*). Algae are extensively distributed in extreme environments, such as deserts, craters, salt lakes, and polar region (*Brock, 1975*; *Carson & Brown, 1978*; *Rayburn, Mack & Metting, 1982*; *Hu et al., 2003*; *Khalil et al., 2010*). Therefore, the exploration of algal germplasm resources has been a new research focus. Some algae can grow under extreme conditions. For instance, *Dunaliella bardawil* and *Aphanothece halophytica* can survive high-concentration NaCl (*Brock, 1975*; *Khalil et al., 2010*), and *Microcoleus,*

Corresponding author
Qingjie Guan,
guanqingjie@nefu.edu.cn

*Scytonema*, *Schizothrix*, *Desmococcus*, and *Stichococcus* can survive arid and semi-arid deserts (*Hu et al., 2003*). However, the saline soil of the Songnen Plain of Northeast China consists of NaHCO₃ and Na₂CO₃ with pH values ranging from 9.0 to 10.5, and reports on algae growing under carbonate conditions are rare. Thus, in the present study, we have isolated and purified a halophilic and alkaliphilic microalgal strain from extreme carbonate soil (alkali spot). The isolated microalga was subjected to morphological, molecular and halophilic (NaCl, NaHCO₃) characterization. This halophilic and alkaliphilic microalga can be used to improve carbonate soil and explore resistance genes.

## MATERIALS AND METHODS

Soil samples of alkali spot from the saline Songnen Plain of Northeast China (Anda City) were collected. The algae were isolated using the streak plate method. A loop of algae was  streaked in Bold's Basal Medium (BBM) solid medium for inverted culture. When the algal colonies grew, they were again streaked until single colonies appeared on the plate upon microscopic examination. The algal species was stored in the Alkali Soil Natural Environmental Science Center of Northeast Forestry University.

## Methods

### The culture of microalgae

The microalga was cultured on improved BBM medium (*Fabregas et al., 2000*) consisting of 0.250 g/L NaNO₃, 0.025 g/L NaCl, 0.075 g/L K₂HPO₄, 0.175 g/L KH₂PO₄, 0.011 g/L H₃BO₃, 0.050 g/L EDTA, 0.031 g/L KOH, 0.490 mg/L Co(NO₃)₂ · 6H₂O, one mg/L CuSO₄, 1.440 mg/L MnCl₂ · 4H₂O, 8.820 mg/L ZnSO₄ · 7H₂O, 0.710 mg/L MoO₃, 0.075 g/L MgSO₄ · 7H₂O, 0.025 g/L CaCl₂ · 2H₂O, five mg/L FeSO₄ · 7H₂O, and KOH adjusted pH 8.0.

### Morphological observations

*Optical microscopy*

The algal cells cultured to the logarithmic growth phase were visualized using optical microscopy. Under 4,000× the total magnification, the morphology of intravital algal cells were directly observed.

*Transmission electron microscopy thin sectioning and observation*

Algal cells were collected by centrifugation, and 3% glutaraldehyde fixative (0.1 mol/L PBS, pH 7.2) that was 20 times the volume of algal cells was added and fixed for more than 24 h at room temperature. After being rinsed by 0.1 mol/L PBS, the algal cells were treated with 1% osmic acid-fixed liquid for 4 h at room temperature (0.1 mol/L PBS, pH 7.2), dehydrated with graded ethanol at increasing concentrations (50%, 15′ × 1; 70%, 15′ × 1; 80%, 15′ × 1; 90%, 15′ × 1; and 100%, 15′ × 2), soaked and embedded with Spurr, and then polymerized overnight at 80 °C. The cells were later sliced with ultramicrotome LKB-1, double stained with uranyl acetate and lead citrate, observed, and then photographed under an electron microscope (Hitachi-7650).

### DNA extraction and analysis of sequence data

Improved CTAB method (*Porebski, Bailey & Baum, 1997*) was employed to extract the genomic DNA of microalga SAE1. Algal cells at the exponential growth phase in suspension

liquid were collected by centrifuging at 10,000 rpm for 5 min, grounded to powder in liquid nitrogen with mortar and pestle. The cells were added into a preheated $2\times$ CTAB solution at 65 °C and incubated in bathtub at 65 °C for 30 min. After cooling to room temperature, it was added with equivalent chloroform:isoamyl alcohol (24:1), and the obtained solution was centrifuged at 12,000 rpm, 4 °C for 10 min. The supernatant was placed into a new centrifuge tube, to which equivalent isopropanol was added. The mixture was deposited at −20 °C for 30 min and then centrifuged at 12,000 rpm, 4 °C for 10 min. The supernatant fluid was discarded, and the rest was rinsed twice with 75% ethanol and then added with deionized water containing RNAase to dissolve DNA after drying at room temperature.

We amplified the partial 18S rRNA gene regions using the primer sets 18SF: 5′-AACCTGGTTGATCCTGCCAGT-3′ and 18SR:5′-TTGATCCTTCTGCAGGTTCACC-3′ (*Katana et al., 2001*). The PCR reaction conditions were as follows: 5 min at 94 °C, 30 cycles for 1 min at 94 °C, 1.5 min at 55 °C, 2 min at 72 °C, and a final extension for 10 min at 72 °C. Then, agarose gel electrophoresis and recovery of DNA were performed. We sequenced DNA to BGI (Beijing Genomics Institute, Shenzhen, China). We analyzed the sequences by a similarity search using BLAST software (http://blast.ncbi.nlm.nih.gov/Blast.cgi). We aligned multiple sequences using the ClustalX 1.8 program. 18S rRNA of the microalgae was sequenced, and its relative genetic distance was calculated with MEGA5.0 software. A phylogenetic tree was constructed with the maximum likelihood method (UPGMA), to test the confidence of the tree topologies, we performed bootstrap analyses for maximum likelihood method (1,000 replicates).

### Analysis of the halophilic and alkaliphilic features of the microalga

A one mL aliquot of the microalga SAE1 solution at the logarithmic phase was added into sterile BBM medium. Then, filter-sterilized NaCl and $NaHCO_3$ solution were added. The final concentrations of NaCl were successively 0, 100, 200, 400, 600, 800, and 1,000 mmol/L, whereas those of $NaHCO_3$ were 0, 50, 100, 200, 300, 400, and 500 mmol/L. After mixing, the solution was cultured at 25 °C under a 16:8 h L:D cycle condition, shook several times every day, also observed and recorded their growth after 14 h. The value of $OD_{\lambda = 700}$ was measured by a spectrophotometer.

Monoclonal microalgae on the plate were picked for culture onto BBM liquid medium at 25 °C under a 16:8 h L:D cycle condition. When they grew to the logarithmic phase, NaCl at the final concentration of 200 mmol/L and $NaHCO_3$ of 400 mmol/L were added to stress culture for 2 days. The cellular structure of the microalga was observed via transmission electron microscopy (TEM), and the slice-making method was performed as previously described.

## RESULTS

### Morphology of microalga SAE1

Single colonies that were isolated and purified by the streak plate method with alkali spot soil solution were recorded and preserved as microalga SAE1. The single algal cells that

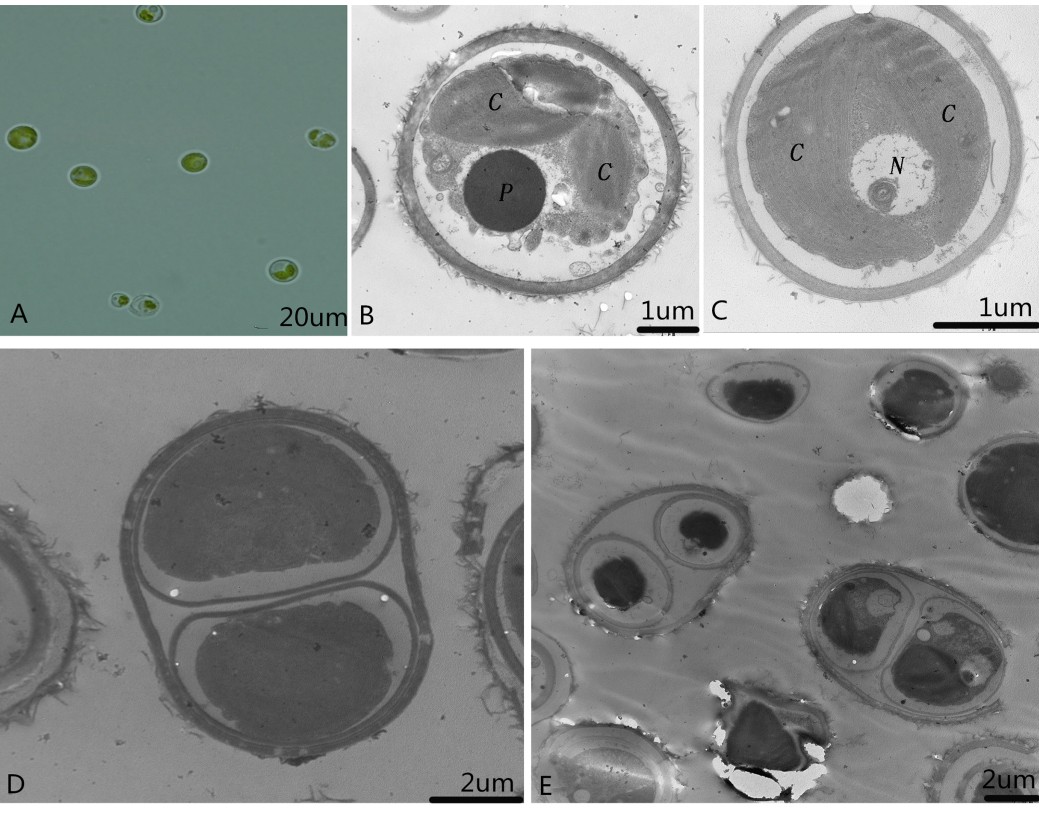

**Figure 1 Morphology of the microalga.** (A) Optical observation of single algal cells; (B and C) microalgal ultrastructure. C, Chloroplasts; N, nucleus; and P, pyrenoid; and, (D and E) microalgal cell division.

appeared green in good growth state were taken for culture in liquid. When the concentration of SAE1 was comparatively low, the solution appeared light green. As the concentration grew, the green color darkened. TEM observation indicates that SAE1 has a simple cellular structure (Fig. 1). It is spherical or nearly spherical with a diameter of four to six μm and a cell wall thickness of 0.22 μm and contains two chloroplasts, one nucleus, one pyrenoid, and several mitochondria, and it reproduces by binary fission.

## Phylogenetic analysis of microalgal SAE1

The length of the 18S rRNA sequence of SAE1 microalgae after PCR amplification, electrophoresis, purification, and sequencing was 1,795 bp. The nucleotide sequence was submitted to GenBank. Homologous comparison with the data of GenBank showed that it has 99% similarity to *Nannochloris* sp. *BLD-15*, with only four base substitutions, and 98% similarity to *Nannochloris* sp. *AS 2-10*. Homologous comparison using the phylogenetic tree (Fig. 2) constructed by MEGA5.0 with the maximum likelihood method shows that SAE1 belongs to the same branch of *Nannochloris* sp. *BLD-15*. Thus, microalga SAE1 was tentatively identified as a species of the genus *Nannochloris*.

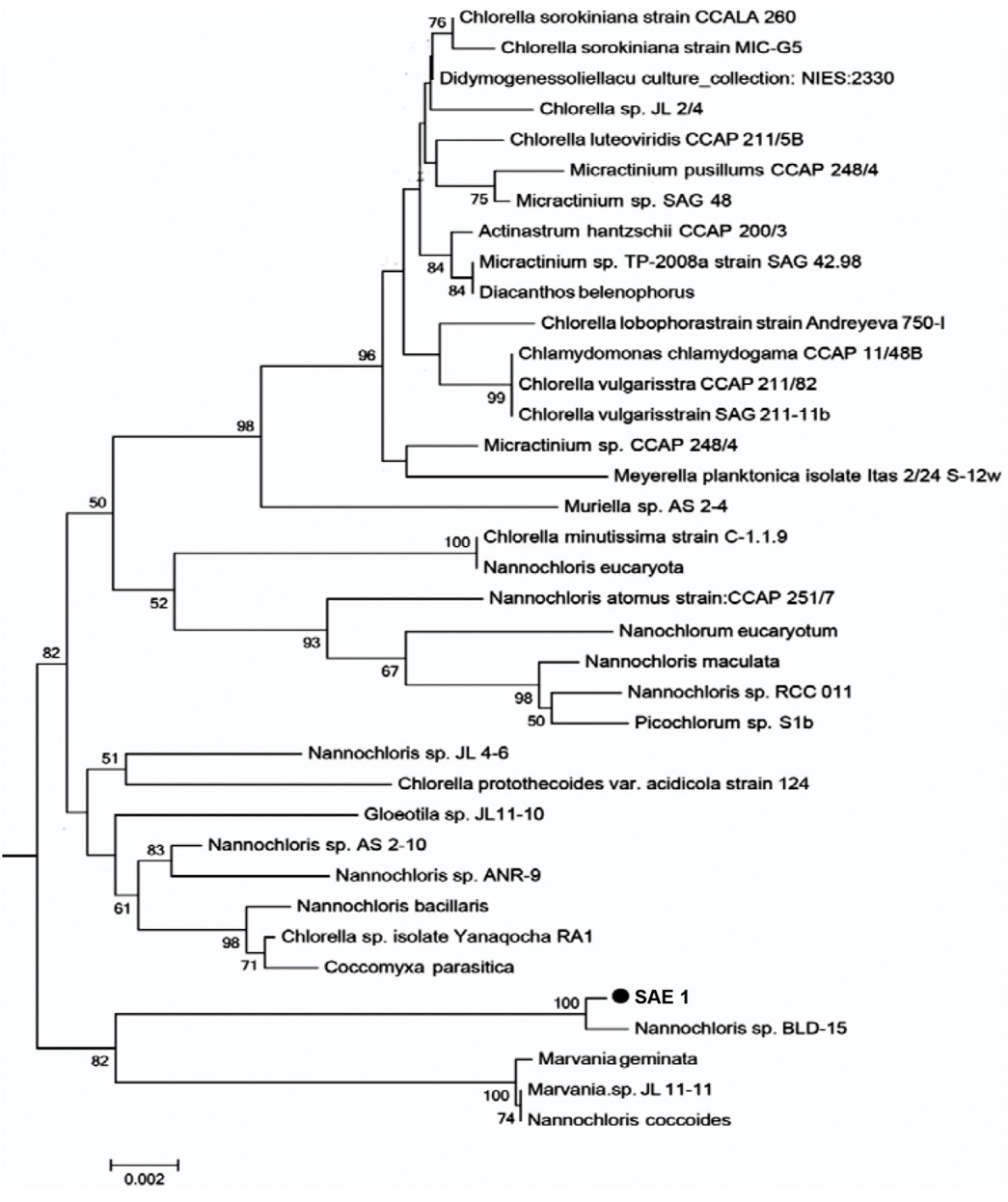

**Figure 2 Molecular phylogeny of the extreme saline–alkali microalgae (SAE1) based on SSU rRNA gene sequence comparisons.** A phylogenetic tree was constructed using the neighbor-joining method. Bootstrap values were calculated 1,000 times, and values below 50% were not included. Sequences were obtained from GenBank (National Center for Biotechnology Information (NCBI)). Results were identified SAE1 as *Nannochloris* sp.

## Salt and alkaline tolerance of microalga SAE1

### Growth pattern of microalga SAE1 in NaCl and NaHCO₃ treatments

The growth-consistent microalgae SAE1 cells were examined for phenotype under $NaHCO_3$ resistance stress (Fig. 3A).The green color of the algal cells darkened as the $NaHCO_3$ concentration in the medium increased, indicating that cell growth significantly accelerated. The microalgal cells had slower growth rate in the medium with NaCl than in

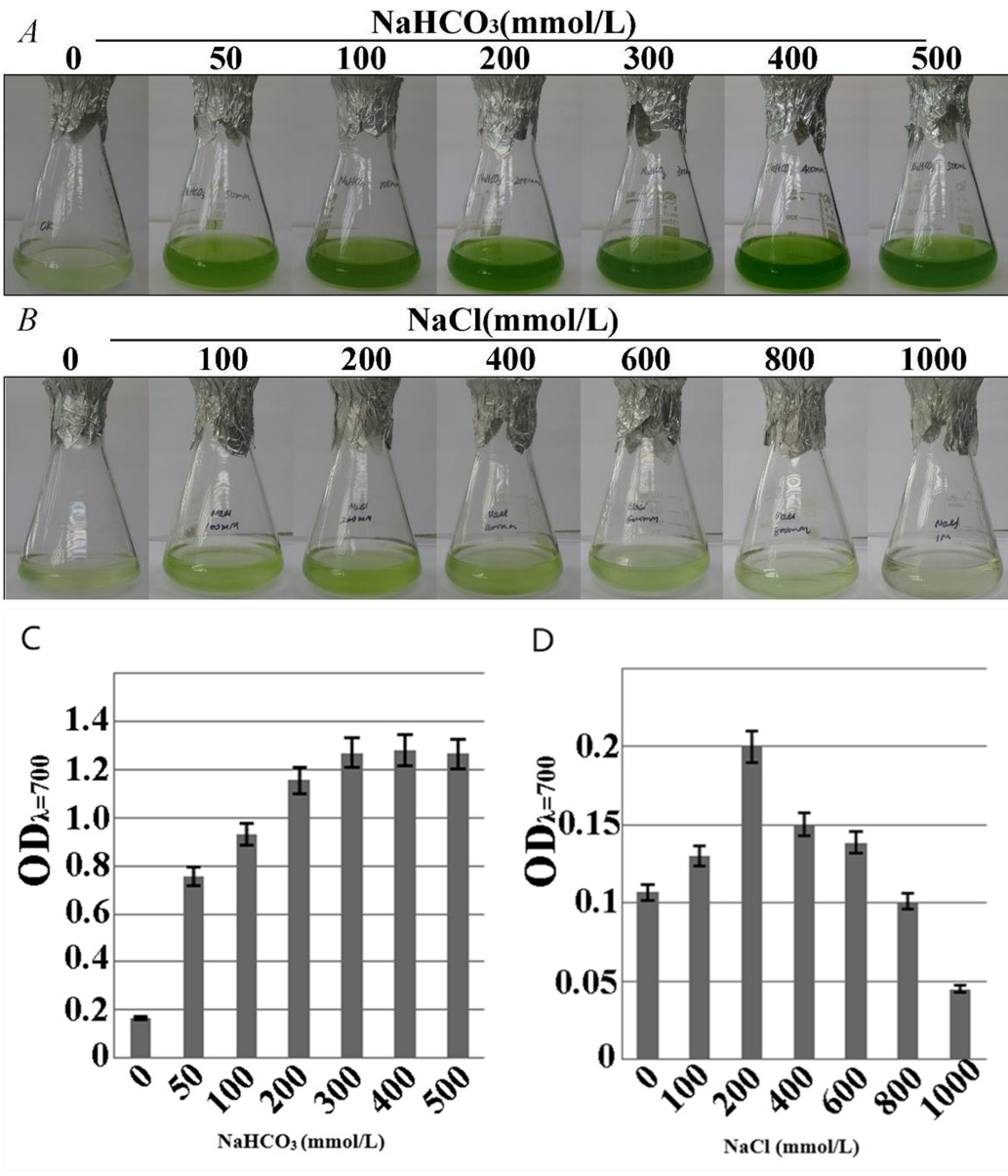

**Figure 3 Growth characteristics of microalga SAE1 under NaCl and NaHCO$_3$ treatments.**
(A) Microalgal growth under different NaHCO$_3$ concentrations; (B) microalgal growth under different NaCl concentrations; (C) microalgal biomass under different NaHCO$_3$ concentrations (OD$_{\lambda = 700}$);
(D) microalgal biomass under different NaCl concentrations (OD$_{\lambda = 700}$).

that with NaHCO$_3$, but appropriate amount of NaCl still promoted the growth of the microalgae. When the NaCl concentration was 200 mmol/L, the phenotypic observation indicated that green color darkened (Fig. 3B). When the NaHCO$_3$ concentration was 400 mmol/L, the growth rate was the greatest (Fig. 3C), as was the value of OD$_{\lambda = 700}$ detected by a UV spectrophotometer, indicating the peak of the exponential growth phase. When the NaCl concentration was 200 mmol/L, the value of OD$_{\lambda = 700}$ was the greatest,
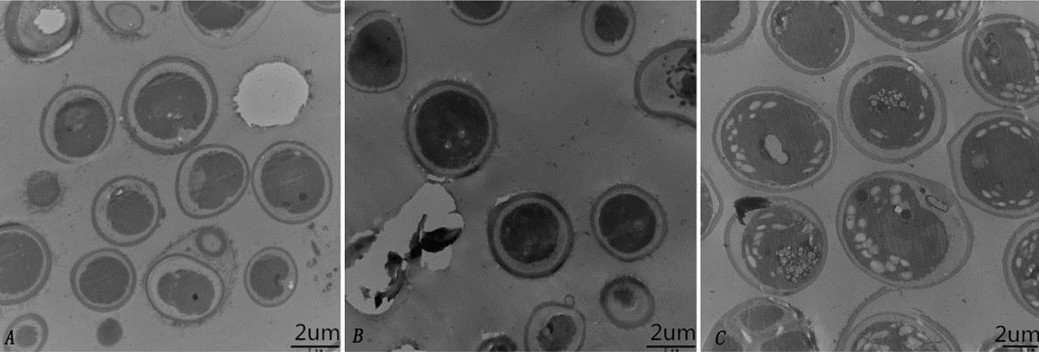

**Figure 4 Changes in microalgal cellular ultrastructure under NaCl and NaHCO₃ treatments.** (A) BBM, (B) BBM with 200 mM NaCl, (C) BBM with 400 mM NaHCO₃. There was no starch grain in the algae cells untreated with saline and alkaline; a few starch grains appeared in three cells under 200 mmol/L NaCl concentration; a great number of starch grains was found in nine cells under 400 mmol/L NaHCO₃ concentration.      

the growth rate of the microalga reached the maximum (Fig. 3D). The cells of SAE1 still survived under the treatment of one M NaCl.

### Growth pattern of microalga SAE1 in NaCl and NaHCO₃ treatments

To further characterize the cellular structure of SAE1 that can grow in high-saline and alkaline environments, We detected by TEM that there was no starch grain in the algae cells untreated with saline and alkaline (Fig. 4A). But only a few starch grains under 200 mmol/L NaCl concentration (Fig. 4B), However, a great number of starch grains was found in the cells under 400 mmol/L NaHCO₃ concentration (Fig. 4C). These results indicate that SAE1 can change intracellular dissolved matter via related metabolism in order to survive.

## DISCUSSION

Salt alga is unicellular green alga belonging to Polyblepharidaceae family. The study of molecular biology in the 21st century steps into the era of genomics or proteomics. *Liska et al. (2004)* adopted 2D protein electrophoresis to compare algal cell proteins growing in low- and high-saline environments and found 76 types of protein that are induced by salt. These upregulated proteins include key enzymes in the Calvin cycle, starch metabolism, and Adenosine Triphosphate (ATP) production in redox; the regulatory factors of protein synthesis and degradation; and an analog of the germ $Na^+$ transporter pump of redox. When saline-tolerant algae migrate from a low-saline to a hypersaline environment, they increase $CO_2$ assimilation and starch degradation by photosynthesis, rendering carbon metabolism, and energy metabolism prone to glycerol synthesis. Thus, a large amount of glycerol is produced, allowing the organism to adjust to the environment (*Liska et al., 2004*). At present, there is little research data on the microbiology of saline–alkali soils. Therefore, it is necessary to study the characteristics of microorganisms in saline–alkali soils and their role in biological survival mechanisms. This will effectively improve future crop cultivation. The soda-salt microalga SAE1 isolated in this study has

important theoretical and practical significance for soil ecological restoration and tolerance-related genetic engineering.

## CONCLUSIONS

The halophilic and alkaliphilic microalga SAE1 isolated in this study has a thick cell wall, different from salt algae without complete cell walls. Analysis of the phylogenetic tree constructed by 18S sequence homologous comparison suggests that SAE1 is highly homologous to *Nannochloris* sp. *BLD-15*. Given its halophilic and alkaliphilic characteristics, SAE1 can survive high $NaHCO_3$ concentrations and high pH values. The amount of intracellular starch grain considerably varies when treated with NaCl and $NaHCO_3$. The saline tolerance of the alga is conferred by the synthesis of abundant glycerine in cells. The primary raw material of glycerol synthesis is starch, which generates dihydroxyacetone phosphate by glycolysis, and later produces glycerine by the catalysis of α-glycerophosphate dehydrogenase and glyceraldehyde 3-phosphate phosphatase (*Goyal, 2007*). One of the alkaliphilic characteristics ($NaHCO_3$) of microalga SAE1 in survival metabolism is the generation of abundant starch grains, which reveals that glycerol synthesis increases osmolality, thereby strengthening its resistance to saline–sodic conditions. The features of alkaliphilic microalga SAE1 determine that it can grow in the carbonate soil of Songnen Plain. The microalga SAE1 will be of great significance to the improvement of carbonate soil and the exploration of resistance genes.

### Funding

This work was supported by the National Key Research and Development Program of China (2016YFC0501203), the Natural Science Foundation of Heilongjiang Province (C2017009), and the Heilongjiang Postdoctoral Scientific Research Developmental Fund (LBH-Q15004). The funders had no role in study design, data collection and analysis, decision to publish, or preparation of the manuscript.

### Grant Disclosures

The following grant information was disclosed by the authors:
National Key Research and Development Program of China: 2016YFC0501203.
Natural Science Foundation of Heilongjiang Province: C2017009.
Heilongjiang Postdoctoral Scientific Research Developmental Fund: LBH-Q15004.

### Competing Interests

The authors declare that they have no competing interests.

### Author Contributions

- Chenxi Liu conceived and designed the experiments, performed the experiments, analyzed the data, prepared figures and/or tables, authored or reviewed drafts of the paper, approved the final draft.

- Jiali Liu performed the experiments, analyzed the data, prepared figures and/or tables, approved the final draft.
- Songmiao Hu prepared figures and/or tables.
- Xin Wang prepared figures and/or tables.
- Xuhui Wang performed the experiments, analyzed the data, contributed reagents/materials/analysis tools, prepared figures and/or tables, approved the final draft.
- Qingjie Guan conceived and designed the experiments, analyzed the data, contributed reagents/materials/analysis tools, prepared figures and/or tables, authored or reviewed drafts of the paper, approved the final draft.

## Data Availability

The raw measurements are available in Dataset S1 and Dataset S2.

## Supplemental Information

Supplemental information for this article can be found online at http://dx.doi.org/10.7717/peerj.7189#supplemental-information.

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
