# Peer review of "Isolation and identification of a halophilic and alkaliphilic microalgal strain"

_PeerJ, doi:10.7717/peerj.7189_

## Round 0.1 · original submission · Major Revisions

Dear Qingjie,

Please find the editorial letter attached, wherein you will see the decision (major revisions required) for your manuscript. Please address the reviewers' suggestions/comments/criticisms thoroughly in your resubmission.

Thanks for your patience,
John Moreau (handling editor)

Reviewer 1 ·

Basic reporting

The authors in the submitted manuscript titled “Isolation and identification of a halophilic and alkaliphilic microalgal strain” presents information about identification of a microalgal strain JB19 isolated from the saline–alkaline soil of Songnen Plain of Northeast China. Isolation and identification of this train I believe will be of considerable interest to researchers in the field of microbiology. I have highlighted some of the issues with the article below. Addressing these issues would significantly improve the quality of the article.
1. Line 158-163: It is customary to mention the figures in the text sequentially as it appears. As such, I suggest that this section is modified so that the figures are mentioned in correct sequence. Observations to Fig 4A should be mentioned first, followed by 4B and then 4C.
2. Line 144-153: Figure 3B is not mentioned in the text. It should be mentioned next to the results discussed in this section. Also, the figures should be mentioned in the correct sequence, figure 3A followed by 3B then 3C and so on.
3. Line 168-169: the two sentences should be rewritten to convey the message clearly to readers.

Experimental design

1. Lines 101-102: it is not clear what the authors meant by “DNA recovery to BGI to sequence were performed”. Please explain clearly in the text.
2. Line 75: Please mention the total magnification used for the observation instead of the objective lens magnifications only.
3. Line 89: Please mention the method used to ground the algal cells for e.g. by mortar pestle etc.

Validity of the findings

1. Line 176-177: “Thus, a large amount of glycerol is produced…”. I do not think that data presented can definitely conclude this statement. If it is a speculation, then it should be clearly mentioned. On other hand, if it is known from previously published work then it should be clearly referenced and cited.
2. Figure 3: Please mention the concentration units in the x-axis of the bar graphs
3. Figure 4: in the figure legend “NaHCO 3” should be “NaHCO3”
4. The raw data for figures 3C-3D and S2 should be provided in an easily accessible excel or pdf file (or other appropriate format) which could be accessed by readers more easily.

Reviewer 2 ·

Basic reporting

The manuscript entitled "Isolation and identification of a halophilic and
alkaliphilic microalgal strain" by Chenxi Liu et al. identified a new halophilic and alkaliphilic microalgal strain JB19 strain from the saline–alkaline soil of Northeast China.

I have found many issues with this manuscript thus I do not recommend it for publication in its present form.

The English of the manuscript is poor. Sentences and informal and spoken language has been used. Authors are required to rewrite the manuscript in formal scientific language. There are dozens of spaces are missing especially in figure legend and references, also many times it is not easy to find out the meaning of the sentences. I would suggest seeking help of an academic expert with good language skills.

Introduction, results, and discussion sections are poorly written. Due to this, the major impact of this manuscript is hidden.

Major concern: In the earlier publication, Qiao et al. (2015) identified the algal strain JB19. They performed the 18S rRNA gene sequence of this strain, did evaluate the phylogeny, but didn’t characterize further without any explanation. However, in the current manuscript Chenxi Liu et al. performed the same line of experiment, as done by Qiao et al. (2015) for other strain, and checked the salt and alkali tolerance of this strain. So, the isolation of this strain was not performed by Chenxi Liu et al. and hence should be removed from the title of this manuscript.

In the earlier report of Qiao et al. (2015), this strain JB19 group together with Chlorococcum sp and not with Nannochloris sp. But in current manuscript JB19 aligns along Nannochloris sp. can author explain this discrepancy??

Experimental design

Experimental design is repetitive but as per the field and satisfactory, however the manuscript is poorly written, due to this the major impact of this manuscript is hidden. There are certain experiments needed that I had mentioned in general comments for the authors.

Validity of the findings

The manuscript entitled "Isolation and identification of a halophilic and
alkaliphilic microalgal strain" by Chenxi Liu et al. claim to identify a new halophilic and alkaliphilic microalgal strain JB19 from the saline–alkaline soil of Northeast China.
The strain was isolated and phyletic history was determined earlier, which is different in current report.

Additional comments

Line 21-23: Rewrite the sentence. “Morphological observation revealed that JB19 has a simple cellular structure; it is unicellular and spheric with a diameter of 4–6 μm and a cell wall thickness of 0.22 μm and contains two chloroplasts and one nucleus.”

Line 24: “constructed by 18S sequence homologous comparison” should be “constructed by 18S sequence homology”.

Line 41: References are not up to date. Need more recent references.
Irregular spacing in the references throughout the manuscript.

Line 45-46: “salt lakes, and north and south poles” to “salt lakes, and polar region”

Line 53-54: ‘we isolated and” to “we had isolated”

Line 55-56: “The microalga was subjected to morphological observation, molecular level identification, and halophilic (NaCl, NaHCO3) characteristic analysis.” Verb repletion in this sentence.
Can be rewritten as- “The isolated microalga was subjected to morphological, molecular and halophilic (NaCl, NaHCO3) characterization.”

Line 59: subheading “1.1 Materials” is not appropriate for this section. Here you have described a full methodology.

Line 62: “A loop of algae was taken with an inoculating loop and then streaked” to “A loop of algae was streaked”

Line 62-64: “When the algal colonies grew, the algae were again streaked until single colonies appeared on the plate upon microscopic examination.”
Authors are required to use the formal scientific language throughout the manuscript.

Line 74: “detected” to visualized

Line 76: “cells was” to “cells were”

Line 77: “Transmission electron microscopy slice-making and observation”
Change the subheading. “Slice –making” is not an appropriate word instead use “thin sectioning”.

Line 78: “and 3% glutaraldehyde fixed liquid” what does this mean?? Liquid was fixed by GA??
Line 86: “1.2.3 Systematic evolution of microalga JB19”. What is the meaning of systematic evolution here in method section???

Line 89: “ground” to “grounded”

Line 90: “bathed” to “incubated in bathtub”

Line 91: “the cells were” it is no longer an intact cells, it has turned into lysate.

Line 92: “supernatant fluid was” to “supernatant was”

Line 98: The sentence is unclear and like that many sentences further in Material and Metods section are ambigious, unclear and grammatically incorrect like: “The primers were universal to eukaryotic algae: 18SF: 5′ AACCTGGTTGATCCTGCCAGT-3′ and 18SR:5′-TTGATCCTTCTGCAGGTTCACC-3′ (Katana et al.,2001).

Line 101-102: “to BGI to sequence.” To to BGI was performed. Also explain BGI.

Line 102-106: “Homology comparison was carried out on NCBI (http://blast.ncbi.nlm.nih.gov/Blast.cgi). With the software ClustalX(1.8), 18S rRNA of the microalgae was sequenced, and its relative genetic distance was calculated with MEGA5.0 software. A phylogenetic tree was constructed with the maximum likelihood method (UPGMA), and bootstrap analyses (1000 replications) were performed to test the molecular phylogenetic tree.”
Very poor sentence structure, please rewrite the whole paragraph.

Line 109: “NaCl and NaHCO3 solution was added” to “NaCl and NaHCO3 solutions were added.”

Line 109-111: “The final concentrations of NaCl were successively 0, 100, 200, 400, 600, 800, and 1000 mmol/L, whereas those of NaHCO3 were 0, 50, 100, 200, 300, 400, and 500 mmol/L.” for what??

Line 111-113: After mixing, the solution was cultured at 25 ℃ in light/dark (16h/8h) condition, shook several times every day, and observed after 14 h. The value of ODλ=700 was measured by a spectrophotometer.
Authors are advised to use formal scientific writing pattern

Line 114: 1.2.5 Influence of saline environment on microalgal cellular ultrastructure.
Subheading is inappropriate for method section. It can be used for result subheading.

Line 132: 2.2 Construction of microalgal JB19 phylogenetic tree
Subheading is inappropriate for result section. The result section subheading should be focused on outcome. It should not be “construction of phylogenetic tree”, rather than “what you got know from that tree!”

Line 133: The 18S rDNA of JB19 was obtained by sequencing the PCR products purified by electrophoresis by using universal primers.
What does that mean? PCR product was purified by electrophoresis?? It is PCR product was separated by electrophoresis and desired band was gel eluted. Again; primer was used to amplify target sequence by PCR and not to purify PCR product by gel electrophoresis.

Line 136: “similarity to Nannochloris sp. BLD-15, with only four base substitutions,” Did author checked the sequencing chromatogram for these four base substitution? A comparison of BLD15 and JB19 chromatogram for the mismatch region is required in main figure.

Line 139-140: Thus, microalga JB19 was initially identified to be the physiological race of the genus Nannochloris.
Is author mean to say that prior to their work, it was already known that this new strain JB19 was of genus Nannochloris. It is important point and author should make it clear at the very beginning of introduction.

Line 142: 2.3 Growth characteristics of the salt and alkaline tolerance of microalga JB19.
The subheading is repeatitive as also in 2.3.1. It may be changed to “salt and alkaline tolerance of microalga JB19”.
Then for follow up subheading: 2.3.1 would be Growth pattern of microalga JB19 in NaCl and NaHCO3 treatments, and
2.3.2. Morphological changes of microalga JB19 under NaCl and NaHCO3 treatments

Line 144: Microalgal JB19 cells in the same growth state were phenotypically observed under resistance to NaHCO3 (Figure 3-A).
What do you mean by this sentence???

Line 145-146: The green color of the algal cells darkened as the NaHCO3 concentration in the medium increased, indicating that cell growth significantly accelerated??
To validate this statement, authors are required to monitor the chlorophyll absorbance as they measured the cell density increase and with respect to NaHCO3 concentration and present the data in bar or line graph.

Line 151: “Phenotype” to “phenotypic”

Line 152-153: The growth rate of the microalga reached the maximum (Figure 3-D) and the cells of JB19 still survived under the treatment of 1 M NaCl.
What is the meaning of this sentence? After growth reached maximum, then cells were put on salt treatment??

Line 156: 2.3.2 Characteristics of the cellular structure of microalga JB19 under NaCl and NaHCO3 treatments.

Line 159: “We detected microalgal cells under 400 mmol/L NaHCO3” to “We analyzed microalgal cells at the concentration of 400 mmol/L NaHCO3”.

Line 160: “Results showed a great number of starch grains” to “Images shows the high accumulation of starch granules”

Discussion needs to be rewritten. Discussion should include the outcome and importance of figures and finding??
In figure um and not uM. Figure legends need to be re-written, with more details. Also author can use of arrow or other symbol to highlight the things they want to explain in figure.

Figure 3 C and D: mention what is on the x and y axis with unit. Also why C and D written in bold red??
In figure 4: please explain in legend about the figure and observation, provide statistical data of how many of these cells showed the morphological changes and granule accumulation?

---

## Round 0.2 · Major Revisions

Dear author as new editor of your manuscript I have found very uusual that suddenly you change the name of the strain that you used in your study from the first version of your manuscript in which your study was based in the same strain name from a previously published one.
I see no clarity in your letter of response and further explanations should be clearly provided. Please explain this unusual situation in the letter of response and mention in text about what and how SAE1 is different from previous JB19 isolate/or if it is same? Also, there are additional comments that need further clarifications since the letter of response has not provided enough arguments for some questions raised by the reviewers.

Reviewer 1 ·

Basic reporting

Authors have addressed all the criticisms

Experimental design

Authors have addressed all the issues which were raised with respect to experimental design

Validity of the findings

Authors have addressed all the criticisms about study

Reviewer 2 ·

Basic reporting

Manuscript has significantly improved after revision but still need minor corrections before acceptance. Rebuttal letter is poorly written and hard to follow. Author should also mention the changes made in manuscript in their "response to reviewer".

Experimental design

Satisfactory.

Validity of the findings

Satisfactory

Additional comments

"The microalgae JB19 in this paper is now changed to microalgae SAE1. The SAE1 strain was re-isolated from the saline-alkali soil in Northeast China and identified a new halophilic alkalophilic microalgae belonging to Nannochloris sp."

Author's explanation is not satisfactory in the above sentence! Still, it is not clear that the JB19 strain from earlier work "Qiao et al. (2015)" is the same strain here? What is the rationale behind changing the name of strain if it is same as mentioned earlier? Please clearly mention in text about what and how SAE1 is different from previous JB19 isolates/ or if it is same?

---

## Round 0.3 · accepted · Accept

Thanks for clarifying the points raised by the reviewers. The manuscript is ready to be accepted after you proofread it carefully. There are many typos in the manuscript (e.g., You need a space before each bracket).

Reviewer 2 ·

Basic reporting

The manuscript entitled "Isolation and identification of a halophilic and
alkaliphilic microalgal strain" by Chenxi Liu et al. has improved after revision. There are minor corrections suggested below after which it is acceptable for publication.

Line 46: polar region.(Brock,1975;Carson and Brown,1978;Rayburn,Mack & Metting,1982;Hu et
Check punctuation and space.

Line 101: BGI(Beijing Genomics Institute, China)

Space missing before bracket, check this type of typos in the manuscript.

Did not need to attach English language certificate in manuscript.

Experimental design

Explained earlier.

Validity of the findings

Explained earlier.

Additional comments

Accepted after minor revision.